# The Molecular Basis of COVID-19 Pathogenesis, Conventional and Nanomedicine Therapy

**DOI:** 10.3390/ijms22115438

**Published:** 2021-05-21

**Authors:** Shirin Kouhpayeh, Laleh Shariati, Maryam Boshtam, Ilnaz Rahimmanesh, Mina Mirian, Yasaman Esmaeili, Malihe Najaflu, Negar Khanahmad, Mehrdad Zeinalian, Maria Trovato, Franklin R Tay, Hossein Khanahmad, Pooyan Makvandi

**Affiliations:** 1Erythron Genetics and Pathobiology Laboratory, Department of Immunology, Isfahan 8164776351, Iran; shirin_ake@yahoo.com; 2Department of Biomaterials, Nanotechnology and Tissue Engineering, School of Advanced Technologies in Medicine, Isfahan University of Medical Sciences, Isfahan 8174673461, Iran; l.shariati@amt.mui.ac.ir; 3Biosensor Research Center, School of Advanced Technologies in Medicine, Isfahan University of Medical Sciences, Isfahan 8174673461, Iran; esmaeili161@gmail.com; 4Isfahan Cardiovascular Research Center, Cardiovascular Research Institute, Isfahan University of Medical Sciences, Isfahan 8158388994, Iran; maryamboshtam@gmail.com; 5Applied Physiology Research Center, Cardiovascular Research Institute, Isfahan University of Medical Sciences, Isfahan 8174673461, Iran; ilnazrahimmanesh@gmail.com; 6Department of Pharmaceutical Biotechnology, School of Pharmacy and Pharmaceutical Science, Isfahan University of Medical Sciences, Isfahan 8174673461, Iran; mina.mirian@pharm.mui.ac.ir; 7Department of Genetics and Molecular Biology, School of Medicine, Isfahan University of Medical Sciences, Isfahan 8174673461, Iran; malih.najaflu@gmail.com (M.N.); zeinalianmehrdad@gmail.com (M.Z.); 8School of Medicine, Isfahan University of Medical Sciences, Isfahan 817467346, Iran; negar_khanahmad@yahoo.com; 9Institute of Biochemistry and Cell Biology (IBBC), National Research Council (CNR), 80131 Naples, Italy; maria.trovato@ibbc.cnr.it; 10The Graduate School, Augusta University, Augusta, GA 30912, USA; ftay@augusta.edu; 11Istituto Italiano di Tecnologia, Centre for Materials Interface, viale Rinaldo Piaggio 34, 56025 Pisa, Italy

**Keywords:** clinical manifestations, coronavirus, COVID-19, oxidative stress, PARP, PARG, RAS pathway, SARS-CoV-2, TRPM2

## Abstract

In late 2019, a new member of the Coronaviridae family, officially designated as “severe acute respiratory syndrome coronavirus 2” (SARS-CoV-2), emerged and spread rapidly. The Coronavirus Disease-19 (COVID-19) outbreak was accompanied by a high rate of morbidity and mortality worldwide and was declared a pandemic by the World Health Organization in March 2020. Within the Coronaviridae family, SARS-CoV-2 is considered to be the third most highly pathogenic virus that infects humans, following the severe acute respiratory syndrome coronavirus (SARS-CoV) and the Middle East respiratory syndrome coronavirus (MERS-CoV). Four major mechanisms are thought to be involved in COVID-19 pathogenesis, including the activation of the renin-angiotensin system (RAS) signaling pathway, oxidative stress and cell death, cytokine storm, and endothelial dysfunction. Following virus entry and RAS activation, acute respiratory distress syndrome develops with an oxidative/nitrosative burst. The DNA damage induced by oxidative stress activates poly ADP-ribose polymerase-1 (PARP-1), viral macrodomain of non-structural protein 3, poly (ADP-ribose) glycohydrolase (PARG), and transient receptor potential melastatin type 2 (TRPM2) channel in a sequential manner which results in cell apoptosis or necrosis. In this review, blockers of angiotensin II receptor and/or PARP, PARG, and TRPM2, including vitamin D3, trehalose, tannins, flufenamic and mefenamic acid, and losartan, have been investigated for inhibiting RAS activation and quenching oxidative burst. Moreover, the application of organic and inorganic nanoparticles, including liposomes, dendrimers, quantum dots, and iron oxides, as therapeutic agents for SARS-CoV-2 were fully reviewed. In the present review, the clinical manifestations of COVID-19 are explained by focusing on molecular mechanisms. Potential therapeutic targets, including the RAS signaling pathway, PARP, PARG, and TRPM2, are also discussed in depth.

## 1. Introduction

Coronavirus Disease-2019 (COVID-19), caused by the Severe Acute Respiratory Syndrome-CoronaVirus-2 (SARS-CoV-2), is a highly contagious disease. The disease was declared a global public health threat by the World Health Organization on March 11, 2020 [1,2]. The virus SARS-CoV-2 is a member of the Coronaviridae family that presently hosts 46 species. SARS-CoV-2 is the third most highly pathogenic virus that infects humans, following the severe acute respiratory syndrome coronavirus (SARS-CoV) and the Middle East respiratory syndrome coronavirus (MERS-CoV). These three coronaviruses are all of zoonotic origin and responsible for life-threatening disease outbreaks in the 21st century [3]. SARS-CoV-2 is a positive-sense single-stranded ribonucleic acid (RNA) virus. It is the newest member the β-coronavirus genus together with SARS-CoV and MERS-CoV. Based on sequence identity [4], the genomic sequence of SARS-CoV-2 is ~79% similar to that of SARS-CoV. Apart from SARS-CoV, MERS-CoV, and SARS-CoV-2, other coronaviruses such as *Gammacoronavirus*, *Deltacoronavirus,* and *Alphacoronavirus* also infect birds and other mammals [5].

To date, there are several vaccines approved for immunization against SARS-CoV-2 and other vaccines are under clinical trials worldwide. Based on various technologies recruited for vaccine development, vaccines are divided into six major categories including inactivated, recombinant spike protein, viral vector, RNA, live attenuated, and virus-like particle vaccines [6]. The efficacy of all registered vaccines depends on testee age, doses of different vaccines, vaccine type and dissimilarity in vaccination procedure. Above all, mutations that occur in the viral genome of SARS-CoV-2 cause controversy regarding the efficacy of the present approved vaccines [7].

It appears an in-depth understanding of the molecular mechanisms involved in COVID-19 pathogenesis is required for better drug and vaccine design. Accordingly, the molecular mechanisms involved in COVID-19 pathogenesis are presented in the present review. Potential therapeutic targets for treating COVID-19 as well as several approved drugs are reviewed. In addition, potential bioactive molecules that may be applicable for the management of the disease are discussed.

## 2. Pathogenesis and Therapeutic Targets

### 2.1. Coronavirus Structure

The genome of coronaviruses (27–32 kb) is a single-stranded positive-sense RNA (+ssRNA). The whole genome sequence of SARS-CoV-2 has been characterized using an RNA-based metagenomic next-generation sequencing approach. The SARS-CoV-2 genome is 29,881 bp in length (GenBank no. MN908947) that encodes 9860 amino acids [8]. The S, E, M, and N genes encode structural proteins, whereas sixteen non-structural proteins (nsp1−16) are encoded by the ORF region [9].

The S glycoprotein is one of the four main structural proteins of SARS-CoV-2, with a size of 180–200 kDa. The S protein is composed of S1 and S2 functional subunits [10]. The S1/S2 protease cleavage site is cleaved by host proteases to activate the protein, which is essential for the fusion of the viral membrane with host cells. The S1 subunit, which consists of the N-terminal domain and the receptor binding domain (RBD), is actively involved in binding to host cell receptors. The S2 subunit mediates viral cell membrane fusion to host cells [9,11].

The initial clinical symptoms of COVID-19 include fever, non-productive cough, nasal congestion, and fatigue. These symptoms are manifested in less than a week after infection [8]. About 75% of patients suffer from severe disease, as seen by computed tomography scan on admission [12]. Pneumonia usually occurs after 10–20 days of the symptomatic infection, which is associated with reduced oxygen saturation, blood gas deviations, and discrete changes in the appearance of the lungs in chest radiographs. Different hematological and inflammatory biomarkers, including lymphopenia, the elevation of C-reactive protein, and pro-inflammatory cytokines, have been used as diagnostic laboratory markers for the diagnosis of COVID-19 [13]. Although SARS-CoV-2 typically causes upper respiratory tract infections, the virus may spread to other tissues such as the central nervous system, cardiovascular system, gastrointestinal system, liver, and kidney, causing damage to those tissues and organs [14,15].

Although scientists discovered that SARS-CoV-2 is genetically similar to SARS-CoV (then named SARS-CoV-1) and MERS-CoV, with about 79% and 50% sequence identity, respectively, the exact mechanism of COVID-19 pathogenesis is still incompletely understood [16]. Homology modeling showed that the RBD structure of the SARS-CoV-2 Spike (S) protein is similar to that of SARS-CoV-1. This suggests that the mechanisms of COVID-19 pathogenesis likely resemble those observed in SARS-CoV-1 infection [16,17].

### 2.2. Coronavirus Life Cycle

Coronaviruses (CoVs) harbor the largest genome among all RNA viruses. The genome is flanked at 5′ and 3′ ends by untranslated regions (UTRs), containing cis-acting secondary RNA elements necessary for RNA synthesis, viral replication, and packaging. At the 5′ end, two large protein-coding open reading frames (ORFs), called ORF1a, and ORF1b, comprising about two-thirds of the viral genome, encode replication–transcription complex (RTC) subunits [18,19]. ORF1a codes for polyprotein (pp) 1a, ORF1a and ORF1b encode cooperatively pp1ab. Both polypeptides are then proteolytically processed into 16 nsps by viral proteases, harboring a papain-like protease (PL-PRO) domain (nsp3) and a 3-C-like protease (3CLpro) domain (nsp5). These nsps assemble into large RTCs that include RNA-dependent RNA polymerase (RdRp), superfamily 1 helicase (HEL1), RNA-processing enzymes, RNA-modifying enzymes, and RNA proofreading function required for the integrity of the CoV genome [18,20]. Among nsps, nsp1 inhibits host gene expression and antiviral signal transduction. Nsp2 interacts with a host protein complex involved in mitochondrial biogenesis, likely disrupting the intracellular signaling. Additionally, nsp2 interacts with nsp3 to form proteases that cleave ORF1ab. Nsp4 is involved in the early secretory pathway in the formation of replication complexes, nsp5 exhibits a protease activity, and nsp6 generates autophagosomes from the endoplasmic reticulum (ER). Nsp7 and nsp8 form a complex with nsp12 carrying the RdRp activity. Nsp9 in complex with nsp8 is involved in viral RNA synthesis and virulence. Nsp13 exhibits an RNA helicase activity, nsp14 in complex with nsp10 carries an exoribonuclease activity, nsp11 and nsp15 are involved in endoribonuclease activity, and nsp16 exhibits a methyltransferase activity [21]. ORFs encoding the major structural proteins (S, E, M, and N) of CoV particles and a varying number of “accessory proteins” are transcribed from the 3′-terminal one-third of the viral genome to form subgenomic mRNAs (sg mRNAs) [18,20].

Coronaviruses are believed to enter host cells through two routes. The first route involves their fusion with the plasma membrane of the host, which enables direct delivery of the viral genome into the host’s cytosol. The second route involves receptor-mediated endocytosis [22].

Concerning the first route of cell entry, it is generally perceived that viral entry occurs via the interaction of the S protein with the host cell’s angiotensin-converting enzyme 2 (ACE2) receptor. The latter is also known as COVID-19 specific cellular receptor [23,24]. One-step proteolytic cleavage, mediated by host cell proteases in certain SARS-CoV S protein residues (position S2′), facilitates direct membrane fusion between the virus and the host’s plasma membrane [25]. Previous studies have shown that SARS-CoV cell entry also occurs through the ACE2 receptor [26,27]. In contrast, MERS-CoV utilizes a two-step furin-mediated membrane fusion mechanism for host cell entry and involves interaction with the dipeptidyl peptidase-4 (DPP-4, also known as CD26) receptor [28]. Upon entry into the host cells, the viral RNA genome is translated into two polyproteins and structural proteins within the cytoplasm, leading to the assembly of the viral progeny [29]. Ultimately, budding occurs after the transit of the translated structural proteins through the endoplasmic reticulum-Golgi intermediate compartment and interaction with the newly synthesized N-encapsidated RNA genome [30].

With respect to the second route of host cell entry, it has been suggested that SARS-CoV-2 uses clathrin-mediated endocytosis as the main mechanism for cell entry [31]. Similar to SARS-CoV and MERS-CoV, SARS-CoV-2 may employ multiple pathways to gain entry into the cytosol of host cells [32,33].

### 2.3. The Renin–Angiotensin System (RAS) Signaling Pathway

The role of RAS in maintaining the normal function of the cardiovascular system is well known. Dysfunction of the RAS is associated with many cardiovascular diseases [34]. Upon release into the bloodstream, renin breaks down angiotensinogen into angiotensin I. Angiotensinogen is produced in the liver and is always present in the blood circulation. Angiotensin I (AngI) is inactive and is converted into Angiotensin II (AngII) by angiotensin-converting enzyme (ACE). This enzyme is found in the vascular endothelium of the kidney and lungs [35]. Angiotensin II subsequently binds to angiotensin type-1 and type-2 receptors (AT1R and AT2R). These receptors are found in the heart, blood vessels, kidney, adrenal cortex, lung, circumventricular organs of the brain, basal ganglia, and brainstem. The receptors are involved in the regulation of blood pressure, body fluids, and electrolyte homeostasis [17].

Angiotensin-converting enzyme 2 is an ACE homolog, belonging to the ACE family of dipeptidyl carboxypeptidases which counterbalances the physiological function of ACE. The ACE2 inactivates AngII by catalyzing the elimination of the C-terminal phenylalanine residue to generate Ang (1–7) or changes Ang I to Ang (1–9). The Ang (1–7) peptide potentially exerts vasodilatory action and negatively regulates RAS via the MAS1 oncogene (Mas) receptor [34,36]. Thus, in COVID-19 pathogenesis, ACE2 acts as a double-edged sword, both as a receptor for viral entry and as a negative regulator for severe symptoms of infection and lung injury [37,38].

Earlier studies of SARS-CoV pathogenesis reported that binding of the SARS-CoV S protein to ACE2 causes a decrease in ACE2. This results in Ang (1–7) downregulation and diminished vasodilation. This process causes excessive production of pro-atrophic, pro-fibrotic, pro-inflammatory, and pro-oxidant agents, thereby exacerbating lung injury and enhancing pulmonary vascular permeability (i.e., wet lung) [39].

The lung provides a massive surface area (~100 m^2^) for viral entry. The alveolar epithelial type II cells (AECII) highly express ACE2 and act as viral reservoirs in the human alveolar compartment [39,40].

### 2.4. Oxidative Stress and Cell Death

Oxidative stress is triggered by the imbalance between production and clearance of reactive oxygen species (ROS). These metabolic by-products of aerobic metabolism include hydrogen peroxide (H_2_O_2_), superoxide radicals (O_2_^−^), singlet oxygen (O_2_), hydroxyl radicals (OH), and peroxynitrite anion (ONOO^−^) [41]. Both viral infection and RAS activation trigger ROS production, resulting in an oxidative burst. The increased ROS levels are responsible for destructive effects on cellular macromolecules such as lipids, proteins and especially nucleic acids such as deoxyribonucleic acid (DNA) [42].

Oxidative stress-mediated DNA damage is repaired via the base excision repair (BER) pathway. This pathway is primarily recruited among the three excision repair pathways (BER, nucleotide excision repair, and DNA mismatch repair) [43]. Normally, poly ADP-ribose polymerase-1 (PARP-1), a DNA base repair enzyme activated by DNA breaks, triggers the BER pathway for the maintenance of genome stability. Upon activation, PARP-1 rapidly utilizes the substrate nicotinamide adenine dinucleotide (NAD^+^) to transfer poly ADP-ribose (PAR) to target damaged DNA [44]. The PARP-1 possesses ADP-ribosyl transferase activity and functions as an antiviral agent through ADP-ribosylation of the viral genome (RNA or DNA) and inhibition of viral transcription. However, several virus families, including Togaviridae, Herpesviridae, and Coronaviridae, encode a macrodomain protein that possesses poly(ADP-ribose) glycohydrolase (PARG) activity. The latter hydrolyzes ADP-ribose units from proteins and nucleic acids to promote optimal replication and virulence [45].

Excessive activation of PARP compensates for ADP-ribose hydrolyzation mediated by PARG. Such activity is associated with catalytic consumption of NAD^+^ followed by adenosine triphosphate (ATP) reduction, leading to depletion of energy and cell death [46]. The DNA damage triggered by PARP and PARG generates a large amount of free ADP-Ribose units which bind to transient receptor potential channel melastatin 2 (TRPM2) through a functional ADP-ribose hydrolase domain in its C terminus [47]. Extracellular calcium concentration, ADP-ribose, and temperature are known to be activators of TRPM2 [47,48]. The TRPM2 present on lysosomal and plasma membranes generates Ca^2+^ influx that results in high Ca^2+^ concentration within the cytosol. The overload of cytosolic Ca^2+^ initiates apoptosis and probably necrosis [49,50]. This is the purported mechanism of severe lung injury in COVID-19 patients [51]. The ability of TRPM2 to respond to oxidative stress made it a promising target for repurposing TRPM2 inhibitor drugs such as fluenamic acid or mefenamic acid, melatonin, and selenium, or for developing new small molecules and therapeutic approaches, such as aptamers [52] and gene editing tools [53,54].

### 2.5. Cytokine Storm

A well-synchronized immune response is essential for the control and eradication of viral infections. Viruses can be detected by the innate immune system. Innate and subsequently adaptive immune responses are activated as host defense [55]. Following SARS-CoV-2 entry into the host cells, the viral RNA can be sensed by pattern-recognition receptors (PRRs) that generate signals for the production of type I interferons (IFNs) and pro-inflammatory cytokines. These molecular mediators help to recruit immune cells such as macrophages to the infection site. Moreover, eosinophilia was proposed to be beneficial in COVID-19 due to its antiviral effect, as previously demonstrated in viral infections e.g., influenza. While eosinopenia is associated with a higher rate of mortality [56].

Together with resident dendritic cells, macrophages are known as professional antigen-presenting cells (APCs). Antigen presentation primes T cells for cytokine synthesis [57]. Viral genomic double-strand RNA activates interferon regulatory factors and the nuclear factor kappa-light-chain-enhancer of activated B cells (NF-κB) pathway, leading to type I IFN synthesis and promotion of pro-inflammatory cytokine responses [58].

An in-silico study demonstrated that the NSP10, S2, and E mRNA of SARS-CoV-2 can potentially bind to Toll-like receptor-3 (TLR3), TLR9, and TLR7 and activate downstream signaling cascades [59]. The dysregulated and aberrant immune responses subsequently result in hyper-inflammation and generate a cytokine storm. This results in multiple organ failure, pulmonary tissue damage, and reduced lung capacity in patients with severe COVID-19 [55,60].

Patients with COVID-19 demonstrate significant increases in the levels of plasma pro-inflammatory cytokines, including monocyte chemoattractant protein 1 (MCP-1), macrophage inflammatory protein -1α, -1β, interleukin (IL) 1-β, interleukin 1 receptor antagonist (IL1RA), IL7, IL8, IL9, IL10, interferon-inducible protein 10, platelet-derived growth factor subunit B, basic fibroblast growth factor, granulocyte colony-stimulating factor, granulocyte-macrophage colony-stimulating factor, interferon-gamma (IFNγ), tumor necrosis factor-alpha (TNFα), and vascular endothelial growth factor A (VEGFA) [61,62].

Overall, the viral infection initiates a detrimental cycle of oxidative stress-mediated dysfunction, including PARP and PARG (macrodomain of the virus and host PARG) activities, ADP-ribose increase, TRPM2 activity, apoptosis and/or necrosis, and release of inflammatory mediators and vasodilators [63]. Together, all these events result in endothelial dysfunction and extravasation of immune cells into the alveolar space, producing a ground glass appearance in chest radiographs. The entrapped immune cells release a high amount of cytokines which leads to systemic inflammatory response syndrome, a scenario usually seen in septic shock and poisoning with paraquat (1, 10-dimethyl-4,40-bipyridinium dichloride). The aforementioned events account for the acute respiratory distress syndrome (ARDS) identified in COVID-10 patients [64,65,66]. Activation of TRPM2 increases NLR family pyrin domain containing 3 (NLRP3) activity and IL8 secretion, intensifying inflammation and cytokine storm [54]. In the inflamed lung tissue, IL8 has a chemotactic role in attracting neutrophils to the lung parenchyma to further exert various pathologic effects.

### 2.6. Endothelial Dysfunction

The earliest and one of the most important indicators of endothelial dysfunction is the chronic reduction of nitric oxide (NO) synthesis and release and/or increased NO degradation by ROS [67]. The reduced bioavailability of NO results in proliferative, pro-oxidative, pro-inflammatory, and pro-thrombotic responses. Various pathological disorders affect endothelial function through changing the molecular mechanisms involved in regulation of NO bioavailability [68].

The optimal environment for the propagation of the coronavirus life cycle is an oxygen-depleted condition. In a hypoxic situation, ROS generation and hypoxia-inducible factor 1-alpha (HIF-1a) activation occur sequentially. These events induce the expression of the furin enzyme, ultimately resulting in cleavage of the spike protein and enabling viral entry into the host cells [69,70].

In the hypoxic milieu, NO release is enhanced by NO synthases through the use of L-arginine amino acid for maintenance of nitroso/redox balance [68,71]. The function of NO synthases is dependent on an adequate amount of tetrahydrobiopterin (BH4) in its active reduced form [72,73]. In the oxidative stress state, extra free radicals interfere with redox homeostasis and adequate BH4 production [74]. It may be concluded that the altered redox homeostasis leads to reduced levels of NO in COVID-19 patients with oxidative burst and RAS activation [75].

Various inflammatory and cardiovascular events occur concomitantly with endothelial dysfunction because of the decreased NO levels. Low levels of NO induce proliferation of vascular smooth muscle cells [76], low-density lipoprotein oxidation [77], as well as the expression of vascular cell adhesion molecule-1 [78] and MCP-1 [79] through inhibition of the NF-κB signaling pathway. Moreover, decreased NO following oxidative burst stimulates the production of matrix metalloproteinases-2 and -9 that contribute to pulmonary damage via degradation of the extracellular matrix [80,81] and induces pro-inflammatory cytokine and chemokine expressions [82]. Platelet aggregation [83], leukocyte adhesion [84], and thrombolysis stimulation [82] are also perceived as manifestations of reduced NO levels that contribute to endothelial dysfunction.

Although the decrease in NO production is prominent in COVID-19 patients, some patients showed a slightly increased level of NO. Nitrous oxide is an important factor in vascular homeostasis because of its role in inhibiting contractile machinery and vasodilatation [85]. In this process, NO produced by endothelial cells spreads to the vascular smooth muscle cells and generates cyclic guanosine-3,5-monophosphate. Subsequently, cyclic guanosine-3,5-monophosphate -dependent protein kinase is activated. The latter contributes to the removal of cytosolic Ca^2+^, which inhibits the contractile machinery, and ultimately vasodilatation [82,86]. Nitrous oxide release in the peripheral vessels may worsen hemodynamic homeostasis and decrease blood pressure and organ perfusion.

Nitrous oxide may also be released from endothelial cells upon shear stress and ischemia. Such events are enhanced by acetylcholine, bradykinin and serotonin [87]. Endothelial dysfunction causes fluid leakage from the vessels to tissues (increased wet/dry ratio and impaired O_2_ transfer in lungs). This, in turn, results in hypovolemia and increased viscosity of the intravascular fluid. High viscosity increases the propensity to thrombosis and micro-infarction in different organs. Hypovolemia and hypotension decrease renal perfusion and cause acute tubular necrosis in the kidney. Cytokines excreted by the kidney usually have a molecular mass between 10 and 22 kDa. With reduced glomerular filtration rate, the low excretion and high production of cytokines lead to the development of a cytokine storm. The summary of the pathogenesis and possible therapeutic targets in COVID-19 patients is illustrated in Figure 1.

## 3. Potential Therapeutics for Management of COVID-19

Different players are involved in the molecular mechanisms of COVID-19 pathogenesis. These players include AgII, ACE2, AT1R, NADPH oxidase, PARP, PARG macrodomain (or NSP3), and TRPM2. These players are perceived by clinical scientists as potential targets for therapy using synthetic drugs or natural compounds.

### 3.1. Vitamin D

Vitamins have been suggested for the treatment of SARS-CoV-2 infection symptoms. Vitamin D is a fat-soluble steroid prohormone with endocrine, paracrine, and autocrine functions [88] with a prescribed daily dose of 2000–5000 international unit (IU). Vitamin D inhibits SARS-CoV-2 through binding to the ACE2 receptor as a mediator of acute lung injury in host cells during the viral infection [89]. The high copy number of ACE2 receptors on the cell surface of human alveolar epithelium considerably facilitates virus internalization and subsequent infection [90]. Experimental administration of the vitamin D agonist, calcitriol, protects the lung from viral injury by modulating components of the RAS pathway, such as ACE I and ACE II, renin, and Ang II [91]. Hence, vitamin D negatively regulates hypertension through direct regulation of RAS activity [92].

Another protective function of vitamin D is based on its immunomodulatory effect via suppression of pro-inflammatory responses and the cytokine storm [93]. Despite the role of vitamin D as a PARP inhibitor, it provides increased amounts of extracellular calcium which facilitates full activation of the TRPM2 channels [94]. Moreover, vitamin D appears to stimulate ACE2 overexpression and may increase viral entry. Nevertheless, the effectiveness of vitamin D in relieving COVID-19 associated symptoms is still controversial and its prescription is limited during the acute phase of COVID-19. Details on the role of vitamin D in COVID-19 pathogenesis are provided in Figure 2.

### 3.2. Thalidomide

Thalidomide, a previously-used nonsteroidal anti-inflammatory drug with the recommended dose of 100 mg QID, was banned in 1961 due to its teratogenic effects. Thalidomide blocks the synthesis of TNF-α, enhances plasma IL-12 levels, peripheral blood CD8^+^ T cells, and INF-γ production [95]. The anti-inflammatory effects of thalidomide on COVID-19 patients helped to improve survival rate and shorten the hospital stay length [6]. Various antiviral properties, including inhibition of PARP, anti-TNF, and -NADPH oxidase have been assigned to thalidomide [96]. The molecular mechanisms of antiviral activity of thalidomide are shown in Figure 3.

### 3.3. Trehalose

Trehalose is a non-reducing disaccharide and is usually used as a stabilizer in drug formulations. The average recommended amount of trehalose is 5 “g”per day. Trehalose affects cellular organelles needed for viral replication and impairs viral function via the autophagy system [97]. Because it inhibits PARP1 and PARP2, trehalose may be used as a PARP inhibitor. The drug has been prescribed for SARS-CoV-2 infected patients through a registered clinical trial [98]. Trehalose down-regulates NF-κB signaling through PARP inhibition. It acts as a chemical chaperon and stabilizes the anti-thrombin protein and has an antithrombotic effect that is similar to heparin. Trehalose induces autophagy via the transcription factor EB by inducing rapid and transient lysosomal enlargement and membrane permeabilization [99] (Figure 4).

### 3.4. N-acetylcysteine

*N*-acetylcysteine (NAC) has been used for the treatment of toxin and drug poisoning with the recommended dose of 1200 mg BID. This is attributed to its PARP inhibitory effect and antioxidant activity by induction of p53 apoptosis. *N*-acetylcysteine may be a suitable candidate for treatment and control of SARS-CoV-2 infection symptoms [96]. *N*-acetylcysteine is a potent scavenger of ROS, hypochlorous acid, and OH [100]. Likewise, NAC has a liver protection effect and may be used as a liver care agent. *N*-acetylcysteine also protects the lung from fibrosis in acute inflammatory conditions such as COVID-19 [101].

### 3.5. Tannins

Tannins are water-soluble polyphenolic compounds and are well known as health-promoting components [102] with a recommended dose of 150 mg BID. Studies have reported the anti-radical activities [102] and anti-inflammatory effects of tannins [103,104]. Based on the molecular mechanisms utilized by SARS-CoV-2 in its pathogenesis, COVID-19 may be considered an inflammatory disorder. As natural antioxidants, tannins can reduce disease morbidity and mortality due to their role in redox homeostasis. A recent study demonstrated that tannins block SARS-CoV-2 by binding to the catalytic residues of 3-chymotrypsin-like cysteine protease (3CL^Pro^) of the virus, which regulates its replication [105]. In addition, the inhibitory effect of tannins on the PARG enzyme has also been reported [63]. Gallotannin is a hydrolyzable tannin that has been shown to inhibit cytokine expression [63].

### 3.6. Flufenamic Acid/Mefenamic Acid and Clotrimazole

Flufenamic acid/mefenamic acid and clotrimazole are potential drugs that inhibit TRPM2 in a non-selective manner and are capable of inhibiting or activating other ion channels [106,107]. Recommended dose of flufenamic acid/mefenamic acid is 250 mg and clotrimazole is 10 mg QID, respectively. Flufenamic suppresses NF-κB nuclear translocation and pro-inflammatory mediator expression. In addition, it induces AMP-activated protein kinase (AMPKα) phosphorylation through the alteration of calcium ion concentrations inside and outside of the mitochondria, ultimately inducing anti-inflammatory action [108].

### 3.7. NAD^+^ and Niacin

Almost all molecular mechanisms involved in COVID-19 pathogenesis are originated from NAD^+^ depletion. Depletion of NAD^+^ is mediated by aberrant PARP activity. This leads indirectly to the reduction of sirtuin 1 activity. Sirtuin 1 deacetylates nuclear proteins, regulates the expression of cytokines and NF-κB, and eventually modulates inflammation, cell survival, and apoptosis using NAD^+^ [109]. One of the consequences of NAD consumption in large amounts is the reduction of ATP levels. This, in turn, results in impairment of all cell activities and loss of cellular integrity. In SARS-CoV-2-mediated ARDS, patients are hypovolemic due to a highly reduced level of aldosterone despite RAS activation. The hypovolemia in COVID-19 patients is due to the lack of serotonin; this biomolecule plays important biological roles including stimulation of aldosterone secretion [110]. Serotonin shortage occurs because of the lack of enough tryptophan as the raw material for serotonin production and NAD synthesis during ARDS. The hypoaldosteronism leads to hyponatremia and eventually hypovolemia in a sequential manner. Fatigue and various degrees of mood disorders are other clinical manifestations of COVID-19 patients that are attributed to consequences of NAD, ATP, and serotonin reduction. In this case, administration of NAD alone may worsen the clinical symptoms due to the high activities of PARP and PARG. Concomitant prescription of NAD, niacin, or its precursor L-tryptophan along with a PARP or PARG inhibitor is a promising strategy against SARS-CoV-2 infection [111,112]. The recommended daily dosage of NAD^+^ is 250 mg and that of niacin is 20 mg.

### 3.8. Losartan

Losartan, the AgII receptor antagonist, reduces the synthesis of TGF-1β and PARP with the recommended dose of 25 mg BID. Hence, losartan appears to be able to prevent or control chronic fibrotic diseases such as cardiac hypertrophy and asthma in addition to reducing hypertension [113,114,115,116]. Losartan therapy improved paraquat-induced pulmonary fibrosis in an animal study. Losartan acts through the inhibition of TGF-1β mRNA expression and synthesis of collagen to prevent pulmonary fibrosis [64].

In addition to its anti-hypertensive effects, losartan significantly reduces platelet aggregation by ristocetin, hematocrit, and hemoglobin level in newly diagnosed hypertensive patients. This confirms the anti-thrombosis and anti-atherosclerosis function of losartan [117].

Another mechanism of action for losartan resides in its immunomodulatory function. The drug significantly regulates the cytokines IFN-γ, IL6, IL17F, and IL22 s in peripheral blood mononuclear cells of rheumatoid arthritis subjects [118].

A recent animal study provided evidence of the positive effects of losartan on severe acute lung injury (ALI) and ARDS (ALI/ARDS). This may be attributed to the inhibition of the NF-κB and the mitogen-activated protein kinase (MAPK) signaling pathways [119]. Activation of the RAS system induces strong oxidative stresses that form the major pathogenic mechanism in COVID19 [120]. Losartan, being an angiotensin receptor 1 (AT1R) blocker in the RAS pathway, may be useful for COVID-19 patients who experience pneumonia [121,122]. Losartan appears to stimulate ACE2 overexpression and may increase viral entry. However, the level of AgII in COVID-19 patients is high and following losartan intake, up-regulation of ACE2 could trigger the protective arm of the RAS pathway.

Based on existing evidence, TRPM2 targeting may be an appropriate strategy for combating COVID-19. There are several small molecules that inhibit TRPM2, including flufenamic acid/mefenamic acid [123,124], 2-(3-methyl phenyl) aminobenzoic acid [123], *N*-(pamylcinnamoyl) anthranilic acid [125], econazole, clotrimazole [126], and 2-aminoethoxydiphenyl borate (2-APB) [127,128]. However, none of these agents are TRPM2 channel-specific [129]. Nucleoside analogs, such as adenosine monophosphate and 8-bromoadenosine 5′-diphosphoribose have also been suggested as potential TRPM2 inhibitors [47,130]. Novel TPRM2 channel inhibitors have been synthesized [106] by modification of the ADP ribose (ADPR) analog, including 8-phenyl-2′-deoxy-ADPR. The latter specifically inhibits TRPM2 without interfering with Ca^2+^ release induced by cyclic ADPR, nicotinic acid adenine dinucleotide phosphate (NAADP), or inositol trisphosphate (IP3) [131]. Melatonin, selenium, and zinc are also TRPM2 inhibitors. The activation of TRPM2 in infected tissues, especially the lungs, causes the influx of extracellular calcium ions into the cytoplasm and promotes apoptosis. Temperature and calcium are TRPM2 stimulators. It is better to limit calcium consumption through nutrition or dietary supplement. Preheating of ventilation air to 37 °C represents the optimum condition for activation of TRPM2 and consequently apoptosis of the infected tissues (such as lungs) that express TRPM2 and contain high levels of ADPR. Because ADPR has a limited activation effect on TRPM2 at 25 °C, it is better to regulate the temperature of the ventilator to 25–30 °C [132]. Because TRPM2 has a pivotal role in COVID-19 pathogenesis, especially in respiratory failure, TRPM2 inhibitors are potential drugs to be valued for combating SARS-CoV-2 infections in human clinical trials.

### 3.9. Remdesivir

Remdesivir is a 1′-cyano-substituted adenosine analog and an RNA-dependent RNA polymerase (RdRp) blocker with arecommended dose of 100 mg per day. Remdesivir is an efficient antiviral agent because of two mechanisms: (i) bond formation with the active site of RdRp [133], and (ii) inhibition of “proof-reading: via the exoribonuclease of SARS-CoV-2. This results in the prevention of viral nucleic acid synthesis [134]. Remdesivir is typically used for treating Ebola virus infection and is currently considered as a potential drug for the management of COVID-19 [135].

### 3.10. Chloroquine and Hydroxychloroquine

Chloroquine and hydroxychloroquine, also known as Ivermectin, has been used against SARS-CoV-2 that minimized viral RNA up to 5000 fold in SARS-CoV-2 infected cells [136]. These are antiviral drugs that recently have been adopted for the treatment of COVID-19 patients with pneumonia [137]. The recommended dose of these bioactive compounds is 250–500 mg BID. It has been postulated that these drugs interact directly with the viruses and hinder their attachment to cell surface receptors [138]. Both drugs act on quinone reductase 2 that is related to the UDP-*N*-acetylglucosamine 2-epimerase structure and the catalysis of sialic acid biosynthesis, being crucial factors for ligand recognition [139]. Chloroquine impairs glycosylation of ACE2, thereby blocking the pre-entry phase of SARS-CoV-2 [138]. In addition, chloroquine impedes SARS-CoV virus replication by increasing the pH of endosomes. Fusion of virus membrane with the host cell membrane occurs at low pH, which facilitates delivery of the SARS-CoV-2 genome into the host cell [140]. An increase in endosomal pH by chloroquine blocks the endocytosis process and disrupts SARS-CoV-2 function. The antiviral mechanisms of hydroxychloroquine and chloroquine are depicted in Figure 5.

### 3.11. Monoclonal Antibodies and Recombinant Proteins

Monoclonal neutralizing antibodies provide passive immunity during virus exposure. Palivizumab is a drug in this category that has been applied for the prevention of respiratory syncytial virus (RSV) infection [142]. Anti-inflammatory antibodies such as anti-IL-6 receptor (Tocilizumab) and anti-ITGA4 (Natalizumab) inhibit inflammation and cellular extravasation [101,102]. Another proposed strategy is the administration of recombinant soluble ACE2 receptors to scavenge the virus. Similar strategies have been used in combating HIV infection via the use of soluble CD4 or CD4-mimicking compounds [143].

### 3.12. Delivery Systems: Role of Nanomedicine

Nanomedicine represents the state-of-art technology for non-invasive diagnosis and targeted treatment of a host of diseases [144]. Because of the unique attributes of nanocarriers in promoting the bioavailability of therapeutics, facilitating cellular uptake, and lowering drug resistance and side effects, nanocarriers are regarded as optimal carriers for diagnosis and treatment of viral infections [145]. To date, a variety of nanocarriers, including lipid-based, polymeric-based, metal-based, and nucleic acid-based carriers have been developed against viral infections [146]. Because multivalent interactions among nanoparticles and viruses are essential for adhesion, recognition, and signaling activation, optimized ligands are required on the nanocarrier surface for effective theranostic applications. Indeed, proper multivalent interactions at the nanoparticle-pathogen interfaces can prevent the pathogens from adhering to the host cell during the early infection stage [147]. Monovalent modifications against viral pathogens are likely to be associated with the rapid development of drug resistance [148]. It is necessary to design and develop nanoscopic antiviral agents that are based on multivalent interactions for effective monitoring of viral particles and inhibiting cell–virus interactions. In the following subsections, potential multivalent theranostic nanoagents that have been experimentally evaluated for the monitoring, diagnosis, and treatment of such viral infections will be discussed.

#### 3.12.1. Organic Nanoparticles

Organic nanoparticles are typically classified as lipid-based nanoparticles, polymer-based nanoparticles, dendrimers, graphene oxide, and cyclodextrins. The predominant advantages of these nanoparticles are their flexibility, easiness of modification, and high stability in biological fluids [149]. Apart from biodegradability and high affinity to specific molecular biomarkers, nanoparticles also exhibit specific characteristics for both diagnosis and treatment [150]. Lipid-based nanoparticles have a high loading capacity for both hydrophilic and hydrophobic drugs, while their outer lipid membrane facilitates cellular uptake of the nanoparticles [151]. The administration of liposomal lactoferrin was useful for the care of patients with SARS-CoV-2 infection [152]. Lactoferrin effectively blocked viral entry through binding to the ACE2 receptors on the host cell membrane. Because of their structure and composition, liposomes are capable of altering the structure of lipids and proteins of viruses [152,153]. Polymers are also useful because of their biocompatibility and easily modifiable features [154,155,156]. Positively charged polymers such as polyethyleneimine, can absorb the viruses with a negative charge, interfere with their genomic structure and inactivate the viruses [157]. Dendrimers can likewise be fabricated in highly-branched 3D structures with the capacity for attachment of multiple functional groups on their surface while encapsulating hydrophobic therapeutics in their core [158]. These exciting properties render nanoparticles suitable for therapeutic applications against tumors, bacterial and viral infections. Because nanoparticles have strong multivalent interactions with viruses, they are highly suitable as antiviral agents for preventing infection of host cells. Cyclodextrin, and its derivatives (α-cyclodextrin, β-cyclodextrin and γ-cyclodextrin) consisting of 6, 7, or 8 α-1,4-linked glucose monomers are the most readily available, commercialized nanoparticles to gain recent attention for antiviral theranostic applications [159]. Interestingly, the hydroxyl groups of cyclodextrin may be replaced to alter its solubility in biological fluids, as well as the binding strength between cyclodextrin and the designated target [160]. Due to the remarkable specific surface area, graphene and its derivatives are promising multivalent 2D platforms with broad theranostic clinical applications [161]. The variety of functional groups on the surface of graphene or graphene oxide enables chemically-selective functionalization with specific target analytes. Graphene oxide can inactivate virus function by direct multivalent interaction of their sharp edges with the virus particles [162]. In addition, the surface of graphene oxide is typically functionalized with other antiviral materials to generate a synergistic effect against the virus. For example, a graphene oxide–based platform has been functionalized with β-cyclodextrin and curcumin to combat RSV (Figure 6A) [163]. This platform inhibits RSV infection of host cells via the prevention of viral attachment, resulting in the inactivation of viral function. Graphene oxide with a high density of sulfates has a high affinity toward viruses. The sulfate groups augment the multivalent interactions between graphene oxide and viruses under physiological conditions, leading to effective antiviral activity (Figure 6B) [164]. Linear polyglycerol azides are similar to the heparan sulfate present in the extracellular matrix. They have better interaction with viruses and have a more potent inhibitory effect when compared to dendritic polyglycerol azides (Figure 6C) [165]. Sulfonates are another category of candidates for functionalization of the graphene oxide surface to suppress the interaction between the viruses and their host cells. After capturing the viruses by the graphene surface, the viral particles may be removed through the use of magnetic nanoparticles, The viruses are subsequently inactivated through exposure to near-infrared radiation (Figure 6D) [166].

Different studies have demonstrated multivalent interactions of organic nanoparticles with immune cells and viral antigens. The results suggest that these nanocarriers are effective immunotherapeutic agents against viral infections, including coronaviruses [159,167,168,169]. Organic nanoparticles have great potential for developing vaccination due to their retention in the systemic circulation. They induce strong immune activity that results in the production of a colossal amount of immunoglobulins [170,171].

#### 3.12.2. Inorganic Nanoparticles

Inorganic nanoparticles such as quantum dots, magnetic nanoparticles, and gold nanoparticles have also been investigated for diagnosis and treatment of various diseases. Inorganic nanoparticles can be non-toxic, hydrophilic, biocompatible, and highly stable when compared with organic materials [172]. Because of their unique chemical, magnetic, and electrical properties, inorganic nanoparticles are highly desirable for diagnostic imaging applications. These applications include luminescence imaging, magnetic resonance imaging, fluorescence imaging, and X-ray computed tomography [173]. The photostability of inorganic nanoparticles enables them to be used for imaging for a prolonged duration. Quantum dots are potential candidates for both diagnosis and inhibition of viruses because of their ease of modification. Quantum dots are capable of binding to the S protein of SARS-CoV-2 to inhibit replication of the viral genome. They also facilitate the monitoring of virus infection via fluorescence biosensing [174]. Iron oxide nanoparticles have been used as agents for clinical trials against SARS-CoV-2. Metal oxide nanoparticles, including TiO_2_, ZnO, copper oxide (Cu_2_O), as well as iron oxide with multifunctional properties are applicable in both antiviral diagnosis and therapy [175]. A recent study that investigated the antiviral and photocatalytic process of TiO_2_ indicates that TiO_2_ nanoparticles kill viruses via ROS generation. This causes impairment of lipid membranes, damage of DNA structure, and virus inactivation (Figure 7A) [176]. Another study reported that pre-treatment of murine lung epithelial cells with zirconia (ZrO_2_) nanoparticles resulted in a significant reduction of influenza A virus replication, stimulation of innate immunity, and overexpression of pro-inflammatory cytokines [177]. This mechanism protected the mice against highly pathogenic avian influenza virus, with potential applications against other viral infections. Other studies have reported the potent antiviral potential of zinc (Zn^2+^), including virus inactivation via inhibition of viral uncoating, viral genome transcription, and viral protein translation (Figure 7B) [178].

Gold nanoparticles have been studied extensively for immunotherapy and antiviral applications [179]. They stimulate the immune system and enhance adaptive immune responses through their internalization by APCs. Polyethylene glycol-conjugated IFNα and hyaluronic acid-modified gold nanoparticles (AuNP/IFNα complex) are stable in human serum with antiviral effects against hepatitis C virus infection (Figure 7C) [180]. The antiviral activity of iron oxide nanoparticles against human immunodeficiency virus (HIV) and influenza virus offered theranostic opportunities for combating coronaviruses [181]. Molecular docking analysis of iron oxide nanoparticles indicated interaction of these nanoparticles (Fe_2_O_3_ and Fe_3_O_4_) with the S protein receptor-binding domain (S1-RBD) of SARS-CoV-2 (Figure 7D) [182].

Silver nanoparticles interact with viral particles via two mechanisms: (i) binding to the outer membrane of the virus, thereby inhibiting the attachment of the virus to the cell receptor; (ii) binding to the nucleic acid (DNA or RNA) of the virus, thereby preventing the replication and transfection of the virus within the cell (Figure 7E) [183]. Silver nanoparticles impair the mitochondrial network and inhibit the influx of antiviral interferon regulatory factors-7 transcription factor into the nucleus of lung epithelial cells [184]. A recent study insinuated that silver nanoparticles may have potential antiviral activity against SARS-CoV-2 [183]. Silica (SiO_2_) nanoparticles have also gained significant attention for their large surface areas, and intrinsic surface reactivity that enables selective chemical functionalization [185].

Taken together, inorganic nanoparticles are potential theranostic candidates for diagnosis and multivalent interactions with immune cells and antiviral therapy. However, fundamental investigations should be performed to translate them into clinical studies. Table 1 provides an overview of the antiviral mechanisms of various multivalent nanomaterials. Table 2 provides an overview of the various therapeutics for COVID-19 that are currently under clinical trials.

## 4. Future Perspective

Acute respiratory distress syndrome is manifested clinically during viral infections such as COVID-19, septic shock, poisoning, and exposure to chemical warfare agents. The prognosis of ARDS is poor and there is no specific treatment for such a condition. This life-threatening lung injury that allows fluid to leak into the lungs generally begins with massive oxidative/nitrosative stress. The subsequent DNA damage activates PARP, endogenous and viral (macrodomain) PARG, and TRMP2 activity, which results in apoptosis, necrosis, and parthanatosis. SARS-CoV-2 expresses multi-domain non-structural protein 3 (NSP3) as a potent extraneous PARG and likely activates the RAS that provides fuel for oxidative stress in this circuit. This detrimental cycle consumes NAD and decreases antioxidant capacity, enhancing inflammation and cytokine release. Based on the aforementioned scheme of SARS-CoV-2 molecular pathogenesis, there are several therapeutic candidates available for COVID-19 treatment. Renin inhibitors such as aliskiren, which suppresses RAS, may also be recommended for COVID-19 treatment. Poly (ADP-ribose) polymerase inhibitors, such as trehalose, olaparib, losartan, vitamin D, and NAC, are other therapeutic options in SARS-CoV-2 infection. Apocynin is an NADPH oxidase inhibitor and may be used to intercept and break the detrimental circuit.

## 5. Conclusions

In conclusion, interruption of the aforementioned detrimental infection molecular pathway may help to convert SARS-CoV-2 infection into a mild viral infection with symptoms analogous to the common cold. The aforementioned drugs and supplements have to be scrutinized for their efficacy via registered clinical trials along with conventional multi-drug regimens and anti-viral therapeutic guidelines. A flowchart of SARS-CoV-2 pathogenesis and the antiviral effects of clinical medications for the management of COVID-19 are illustrated in Figure 8.

## Figures and Tables

**Figure 1 ijms-22-05438-f001:**
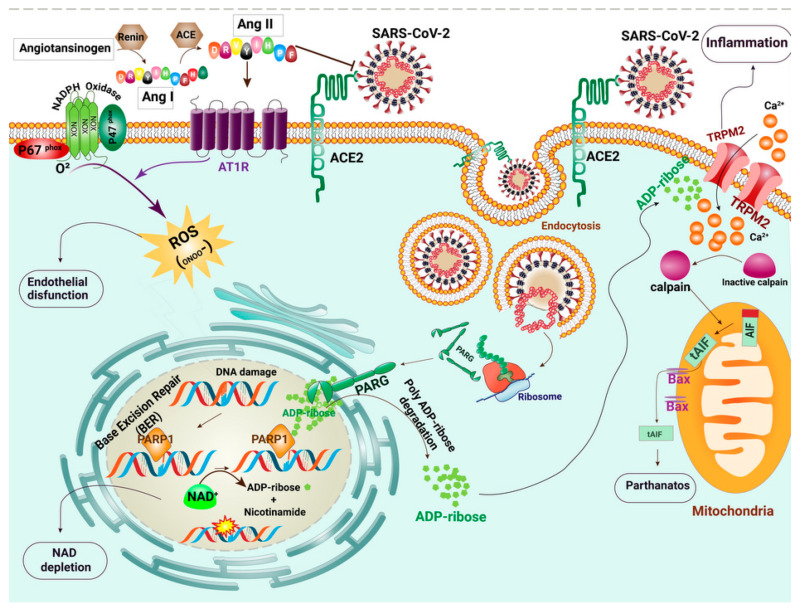
Schematic of the molecular events that occur in SARS-CoV-2 infected cells. SARS-CoV-2 forms a complex with angiotensin-converting enzyme 2 (ACE2) and enters cells via cell membrane-localized receptors. Receptor occupation by SARS-CoV-2 triggers ACE2 down-regulation and renin-angiotensin system (RAS) over-activation. Renin (REN) mediates the transformation of angiotensinogen (AGT) to angiotensin I (Ang-I). The angiotensin-converting enzyme (ACE) metabolizes Ang I to angiotensin II (Ang II). Because of the reduction in available ACE2, the angiotensin type I receptor (AT1R) is activated by the elevated Ang II. Subsequently, AT1R triggers a signaling cascade via activation of NADPH oxidase and induces intense oxidative stress. The produced ROS causes the breakage of DNA strands. Base excision repair (BER) plays a crucial role in repairing oxidative DNA damage via Poly(ADP-ribose) polymerase (PARP) activity. PARP undergoes auto-modification by transferring ADP-ribose on itself. Poly(ADP-ribose) glycohydrolase (PARG) is a protein encoded by Coronaviridae, which separates ADP-ribose units from PAR. Detrimental PARP and PARG activities lead to the accumulation of ADP ribose in the cytosol. Transient receptor potential channel melastatin 2 (TRPM2) channels are over-activated by direct binding of ADP-ribose and release a large amount of calcium ions (Ca^2+^) to the cell. The intracellular Ca^2+^ overload probably results in parthanatos, a form of programmed cell death that is distinct from other cell death processes such as necrosis and apoptosis.

**Figure 2 ijms-22-05438-f002:**
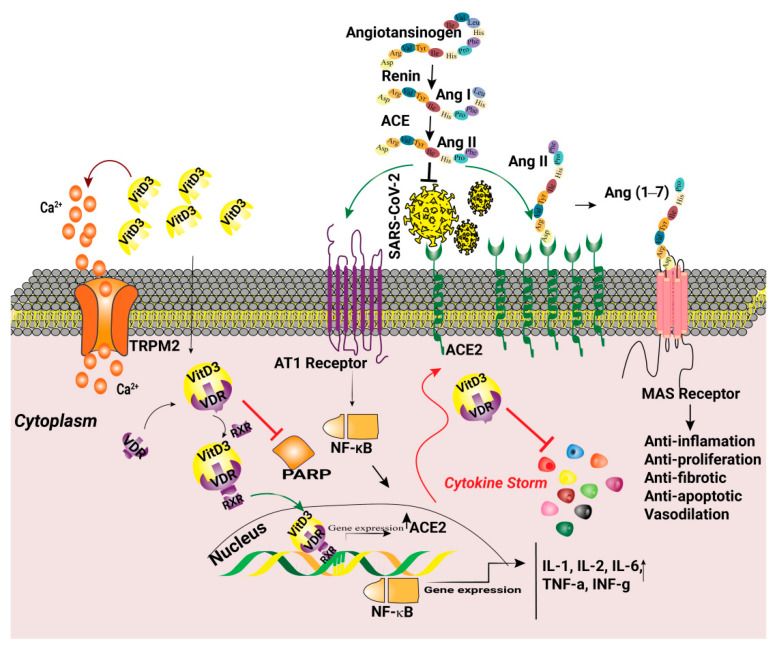
Overactivation of the renin-angiotensin system (RAS) is a consequence of SARS-CoV-2 infection. Binding S protein to the angiotensin-converting enzyme 2 (ACE2) leads to receptor internalization and downregulation and causes upregulation of AngII. AngII binds to AT1R and activates the NFĸB pathway, causing the release of pro-inflammatory cytokines and inducing oxidative stress. Vitamin D3-VDR-RXR complex increases ACE2 and ACE2 augments the generation of angiotensin 1–7 (Ang (1–7)) by metabolizing Ang II. The Mas receptor (MasR) activated by Ang (1–7) induces beneficial actions such as anti-inflammation, anti-proliferation, anti-fibrotic, anti-apoptotic, and vasodilation. Vitamin D3 (VitD3) is also a poly(ADP-ribose) polymerase (PARP) inhibitor. VitD3 increases extracellular Ca^2+^ and results in the activation of transient receptor potential channels, melastatin 2 (TRPM2), and parthanatos.

**Figure 3 ijms-22-05438-f003:**
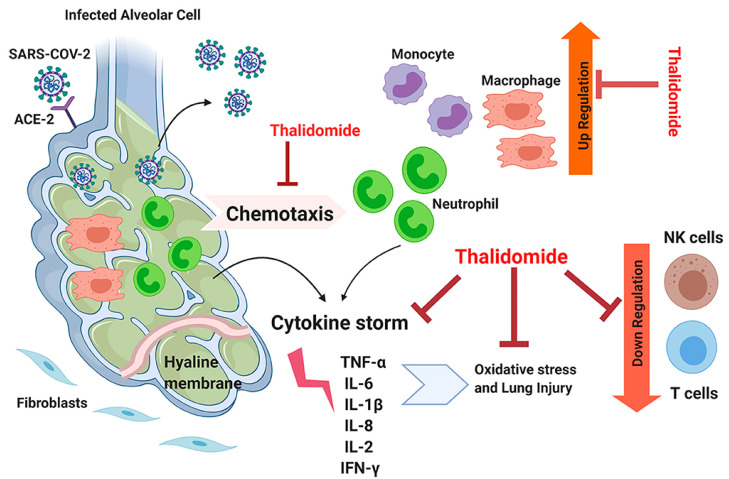
Schematic of the potential molecular mechanisms of the antiviral activity of thalidomide. The drug potentially blocks the chemotaxis of neutrophils and suppresses the cytokine storm and also reduces the generation of oxidative stress. Thalidomide also stimulates natural killer (NK), and T cell activities, resulting in down-regulation of SARS-CoV-2 activities. Reproduced from [88] with permission from Frontiers. Abbreviations: TNF-α, tumor necrosis factor-alpha; IL, interleukin; ACE-2, angiotensin-converting enzyme-2; IFN-γ, interferon-gamma.

**Figure 4 ijms-22-05438-f004:**
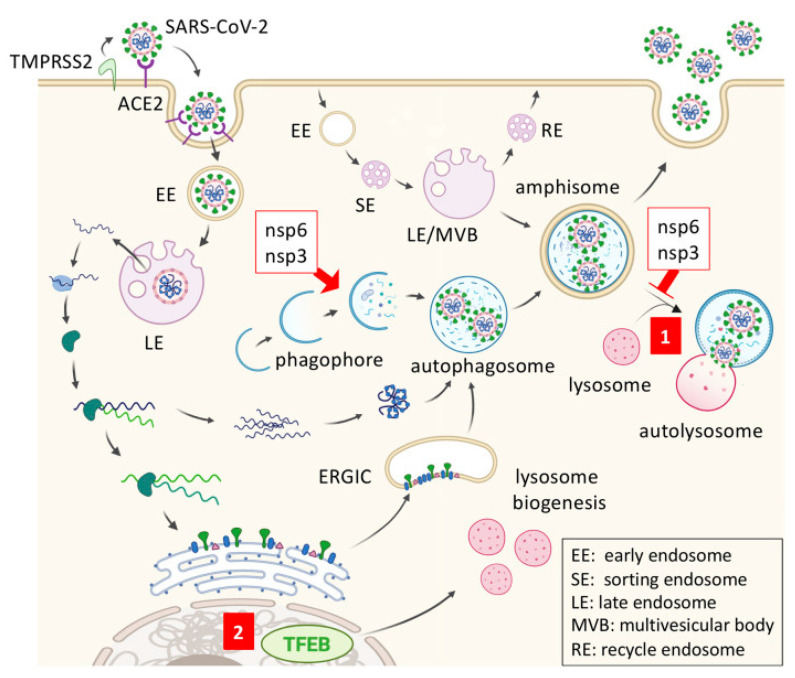
Schematic of antiviral mechanism of trehalose against SARS-CoV-2. SARS-CoV-2 enters the host cell through the interaction of viral spike protein with ACE2 receptor on the cell surface. Coronavirus nonstructural protein 6 (nsp6) and nsp3 promote autophagosome formation but prevent autophagosome-lysosome fusion. (**1**) Trehalose may stimulate amphisome-lysosome fusion for virion degradation after the trafficking of viral double-membrane vesicles, (**2**) Trehalose activates transcription factor EB (TFEB), and up-regulates lysosome biogenesis. Reproduced from [89] with permission from Frontiers.

**Figure 5 ijms-22-05438-f005:**
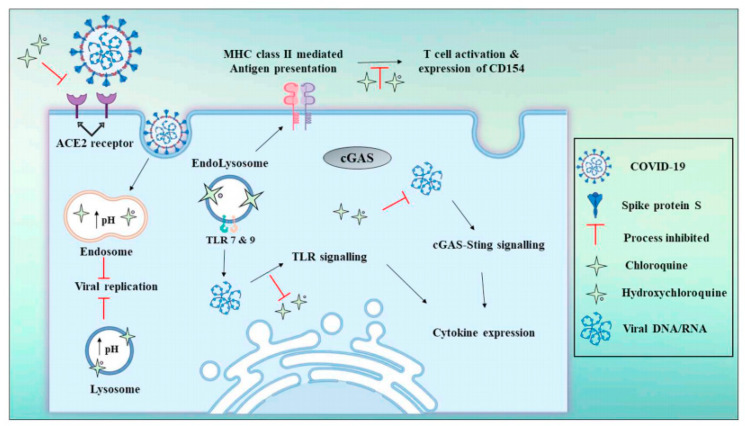
Schematic of the antiviral mechanisms of chloroquine and hydroxychloroquine. These drugs block SARS-CoV-2 invasion as well as reduce the risk of cytokine storm by preventing T cell activation. Reproduced from [141] with permission from Elsevier.

**Figure 6 ijms-22-05438-f006:**
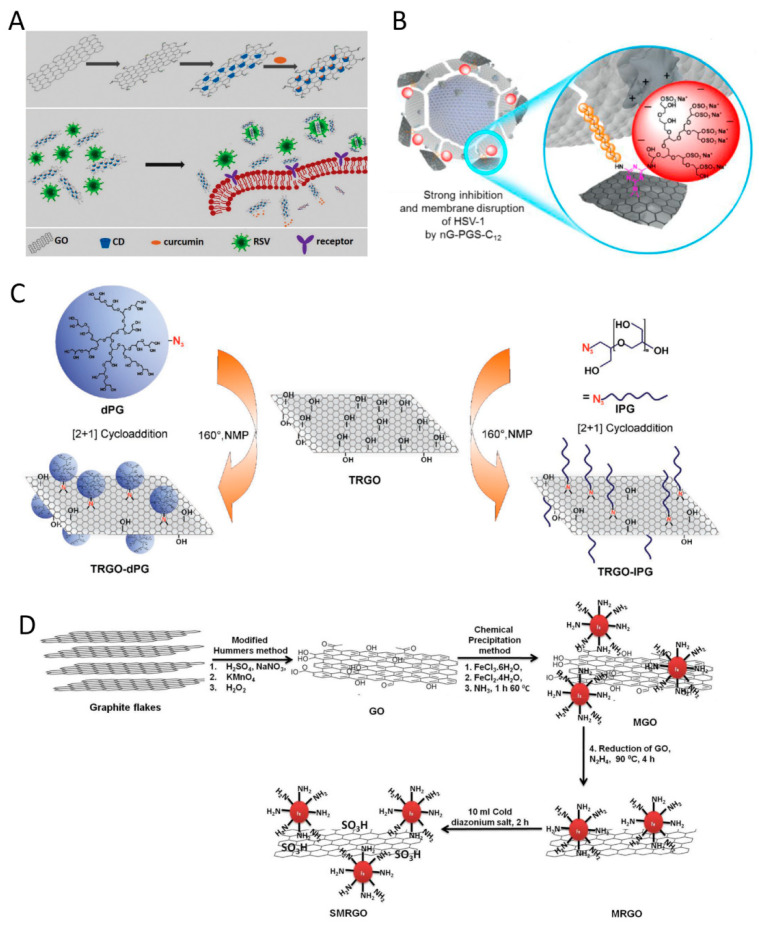
Schematic of various modifications of graphene oxide (GO) for antiviral applications. (**A**) Functionalized GO with cyclodextrin and curcumin to inhibit Respiratory Syncytial Virus (RSV) infection. Reprinted from [163] with permission from the Royal Society of Chemistry; (**B**) Functionalized GO with sulfated groups. Nanographene polyglycerol sulfate coverage enables virus interaction with the positively charged domains of glycoproteins on the surface of Herpes Simplex Virus 1 (HSV-1). Derivatives with long alkyl chain segments (≥C12) show the strongest inhibition and disrupt the virus envelope. Reprinted from [164] with permission from the Royal Society of Chemistry; (**C**) Functionalized reduced GO (rGO) with dendritic polyglycerol (dPG) azide or linear polyglycerol (lPG) azide. Reprinted from [165] with permission from Wiley; (**D**) Functionalization of GO platform with magnetic nanoparticles to produce MRGO. The MRGO is then sulfonated to yield SMRGO. Reprinted from [166] with permission from American Chemical Society.

**Figure 7 ijms-22-05438-f007:**
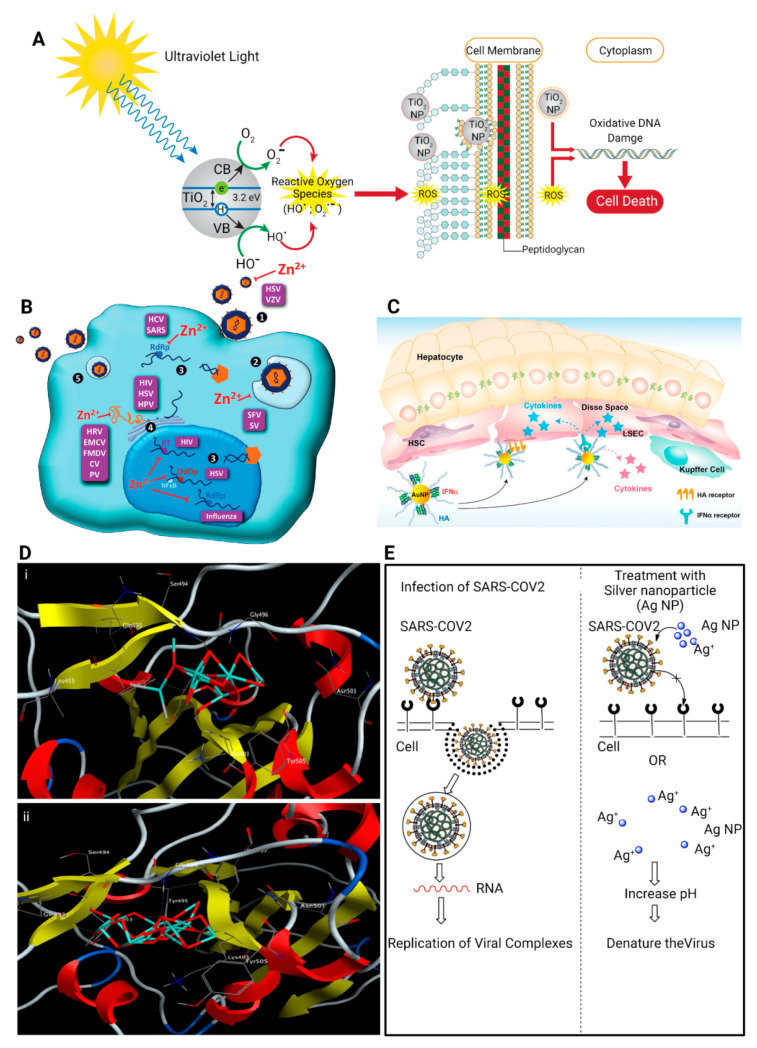
Schematic of the antiviral mechanism of inorganic nanoparticles. (**A**) The antiviral activity of TiO_2_ nanoparticles is attributed to the formation of reactive oxygen species. Reprinted from [176] with permission from Elsevier, (**B**) Antiviral activity of Zn nanoparticles is attributed to the inhibition of viral genome transcription, viral protein translation, and polyprotein processing. Reprinted from [178] with permission from American Society for Nutrients, (**C**) Strategic illustration of HA-AuNP/IFNR complex for the target-specific systemic treatment of hepatitis C infection. Reprinted from [180] with permission from American Chemical Society, (**D**) 3D interaction schematic showing (i) Fe_2_O_3_ and (ii) Fe_3_O_4_ docking interactions with the key amino acids in the receptor- binding domain of the spike protein (S-RBD) of SARS-COV-2. Reprinted from [182] with permission from Elsevier, (**E**) The proposed antiviral mechanism of Ag nanoparticles against SARS-CoV-2. Reprinted from [183] with permission from Multidisciplinary Digital Publishing Institute.

**Figure 8 ijms-22-05438-f008:**
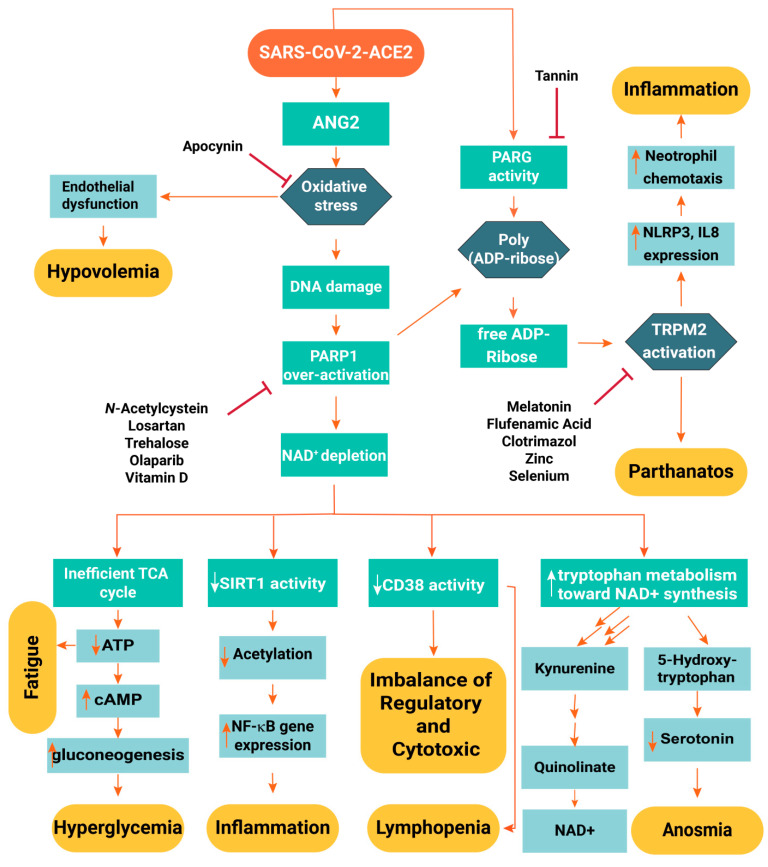
Flowchart of the SARS-CoV-2 pathogenesis and the purported antiviral mechanisms of clinical medications used for combating COVID-19.

**Table 1 ijms-22-05438-t001:** An overview of the antiviral mechanisms of multivalent nanomaterials.

Nanomaterial	Antiviral Mechanism	Targeted Viruses	References
Liposomes	Alter the structure of lipids and proteins of viruses via their composition (fatty acids, surfactants, alcohol)	DENV, SARS-CoV-2, HIV, Influenza	[186,187]
Polymers	Absorb negative charged virus particles via their positive charge and inactivate viruses	HBV, HDV, HSV, Influenza, CoVs	[155,188]
Dendrimers	Strong multivalent interactions with viruses via their highly-branched 3D structure and inactivate viruses	HIV, HCV, SARS-CoV-2	[189,190,191]
Cyclodextrin (CD)	Directly inactivate virus and inhibit attachment	HIV, RSV, HSV-2, DENV-2, ZIKV	[159]
Graphene oxide (GO)	Inactivate virus function because of the interactions between the sharp edges of the GO layers and the virus particles	RSV, CoVs, HPV	[163,192]
Quantum dots (QDs)	Inhibit virus entry at the early stage via interaction with the spike protein of viruses and block genomic replication	SARS-CoV-2, HIV-1	[174]
Gold NPs (AuNPs)	Prevent viral attachment and penetration	DENV, HCV, HPV, CoVs	[193,194]
Iron oxide NPs (IONPs)	Interaction with the spike protein receptor binding domain (S1-RBD) and block virus replication	HIV, Influenza A, SARS-CoV-2	[181,182]
Titanium oxide NPs (TiO_2_NPs)	Generate ROS leading to impair lipid membranes and damage DNA structure of virus particles	CoVs, Influenza A	[195,196]
Zirconium oxide NPs (ZrO_2_NPs)	Stimulation of innate immunity and promote the expression of cytokines	H5N1 influenza virus, Influenza A	[177]
Zinc oxide NPs (ZnONPs)	Releasing of Zn^2+^, generate ROS and block viral replication and translation	HIV, HSV, HPV, H1N1 influenza	[197]
Copper oxide NPs (CuO_2_NPs)	Releasing of Cu^2+^, oxidation of viral proteins and degradation of viral genome	HSV	[198,199]
Silver NPs (AgNPs)	Iron release (Ag^+^), increase ROS and oxidation viral proteins, bind to the nucleic acid (DNA or RNA) of the virus	HSV, SARS-CoV-2	[184,200]
Silicon dioxide NPs (SiO_2_NPs)	Reduce the amount of progeny virus	Influenza virus A	[201]

Abbreviations. Nanoparicles (NP), Respiratory syncytial virus type (RSV), Dengue virus (DENV), Human immunodeficiency viruses (HIV), Hepatitis B virus (HBV), Hepatitis D virus (HDV), Herpes simplex virus type 1 (HSV), Coronaviruses (CoVs), Zika virus (ZIKV), Human papillomavirus (HPV).

**Table 2 ijms-22-05438-t002:** Summary of the drugs for combating COVID-19 that are under clinical trials.

Therapeutic Agent	Current Status	Rout of Administration	Purpose of Study	Clinical Trial Identification
Remdesivir	Phase III	Intravenous	Antiviral drug to manage COVID-19	NCT04678739 NCT04647695 NCT04582266 NCT04738045
Chloroquine	Phase I/II/IV	Oral	Antimalarial drug to treat mild Asymptomatic and Symptomatic cases of COVID-19	NCT04443270 NCT04328493 NCT04627467
Lopinavir/ ritonavir	Phase IV/II	Intravenous	Antiviral drug to treat COVID-19 in terminally sick patients with cancer and immune suppression	NCT04738045 NCT04455958
Azithromycin	Phase III/IV	Oral	Antibiotic drug to treat COVID-19 patients	NCT04365231 NCT04359316
Hydroxy-chloroquine	Phase II/I	Oral	Antimalarial drug to treat ambulatory patients with mild and severe COVID-19	NCT04340544 NCT04351620
Arbidol	Phase IV	Oral	Antiviral drug to treat COVID-19 patients	NCT04350684
Isotretinoin	Phase III	Oral	Retinoid drug to evaluate the safety and efficacy in treatment of COVID-19	NCT04663906 NCT04353180
Lenalidomide	Phase IV	Oral	Antiangiogenic drug to treat mild COVID-19 patients	NCT04361643
Chlorpromazine	Phase III	Oral	Antipsychotics drug to manage COVID-19 subjects	NCT04366739
Canakinumab	Phase III	Intravenous	Anti-human-IL-1β monoclonal antibody to study the efficacy in treating COVID-19 patients	NCT04362813
Ruxolitinib	Phase III	Oral	Antiviral drug to assay the efficacy in COVID-19 patients with cytokine storm	NCT04362137
Dexamethasone	Phase IV	Intravenous	Steroid drug to treat COVID-19 patients	NCT04325061
Favipiravir	Phase III	Oral	Antiviral drug to treat COVID-19 patients	NCT04336904
Sildenafil citrate	Phase III	Oral	Phosphodiesterase inhibitor drug to treat COVID-19 patients	NCT04304313

## Data Availability

In this section, please provide details regarding where data.

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
