# Peer review of "The Molecular Basis of COVID-19 Pathogenesis, Conventional and Nanomedicine Therapy"

_ijms, 2021, doi:10.3390/ijms22115438_

Round 1
Reviewer 1 Report
A very interesting and complete review on COVID pathogenesis, and on all the possible treatments proposed in the management of this condition. The review is very broad and useful in my opinion also for the clinician; I have some queries:
A materials and methods section stating what databases were searched in order to perform this review and what keywords were used in order to extract the content of this paper.
page 4 line 165 you should add "high levels of eosinophils have been linked to a good response to this condition, while low levels are associated to higher mortality" and cite an article such as doi: 10.1111/dth.13681.
page 7 line 235 you should add: "Vitamin D is a fat-soluble prohormone steroid that has endocrine, paracrine and autocrine functions. " and cite an article such as doi: 10.1007/s13668-020-00322-4.
Thank You
Author Response
AUTHORS’ REPLY TO REVIEWERS’ COMMENTS
Title: The molecular basis of COVID-19 pathogenesis, conventional and nanomedicine therapy
Authors: Shirin Kouhpayeh †, Laleh Shariati †, Maryam Boshtam, Ilnaz Rahimmanesh, Mina Mirian, Yasaman Esmaeili, Malihe Najaflu, Negar Khanahmad, Mehrdad Zeinalian, Maria Trovato, Franklin R Tay, Hossein Khanahmad*, and Pooyan Makvandi*
Journal: International journal of Molecular Sciences
Manuscript Reference Number: ijms-1221052
We thank the referees for their useful comments which helped us improve the attached revised manuscript. The manuscript was revised according to the referees’ comments. All changes were highlighted in the revised manuscript as Supporting Information. We hope that our revised manuscript will be acceptable for publication in International journal of Molecular Sciences.
Please find below the replies to the referees’ comments.
…………………………………………………………………………………………………
Point-by-Point Response to the Editors and Reviewer’s Comments.
Referee: 1
Comments to the author
A very interesting and complete review on COVID pathogenesis, and on all the possible treatments proposed in the management of this condition. The review is very broad and useful in my opinion also for the clinician; I have some queries:
A materials and methods section stating what databases were searched in order to perform this review and what keywords were used in order to extract the content of this paper.
Response:
Thank you for this suggestion. We used the National Library of Medicine (MEDLINE) (PubMed); Google Scholar; EMBASE; the Scientific Information Database (SID) and the ISI Web of Knowledge database to search for relevant articles published in this field in English language. Keywords including “COVID-19”, “Coronavirus”, “SARS-CoV” were searched in combination with “pathogenesis”, “oxidative stress”, “PARP”, “PARG”, “TRPM2”, “renin–angiotensin”, “Cytokine storm”, “endothelial dysfunction”, “treatment”, therapy”, “management”, “nanomedicine” and “antibody”.
Page 4 line 165 you should add "high levels of eosinophils have been linked to a good response to this condition, while low levels are associated to higher mortality" and cite an article such as doi: 10.1111/dth.13681.
Response: The requested concept plus the reference were added to page 4, line 163. Moreover, eosinophilia proposed to be beneficial in COVID-19 due to its antiviral effect, as previously demonstrated in viral infections e.g, influenza. While eosinopenia is associated with higher rate of mortality.
Page 7 line 235 you should add: "Vitamin D is a fat-soluble prohormone steroid that has endocrine, paracrine and autocrine functions. " and cite an article such as doi: 10.1007/s13668-020-00322-4.
Response: The brief requested description for vitamin D plus the reference were added to beginning of the vitamin D section, page 7 line 238. Vitamin D is a fat-soluble steroid prohormone with endocrine, paracrine and autocrine functions.
Reviewer 2 Report
I believe is suitable for publication. it's well written and well conducted.
Author Response
AUTHORS’ REPLY TO REVIEWERS’ COMMENTS
Title: The molecular basis of COVID-19 pathogenesis, conventional and nanomedicine therapy
Authors: Shirin Kouhpayeh †, Laleh Shariati †, Maryam Boshtam, Ilnaz Rahimmanesh, Mina Mirian, Yasaman Esmaeili, Malihe Najaflu, Negar Khanahmad, Mehrdad Zeinalian, Maria Trovato, Franklin R Tay, Hossein Khanahmad*, and Pooyan Makvandi*
Journal: International journal of Molecular Sciences
Manuscript Reference Number: ijms-1221052
We thank the referees for their useful comments which helped us improve the attached revised manuscript. The manuscript was revised according to the referees’ comments. All changes were highlighted in the revised manuscript as Supporting Information. We hope that our revised manuscript will be acceptable for publication in International journal of Molecular Sciences.
Please find below the replies to the referees’ comments.
…………………………………………………………………………………………………
Point-by-Point Response to the Editors and Reviewer’s Comments.
Referee: 2
Comments to the author
I believe is suitable for publication. it's well written and well conducted.
Reviewer 3 Report
I suggest some corrections:
- In Abstract is description of human cell death by coronaviruses. Please, remove it, and write in Abstract what conclusions are from this review, mention therapeutic targets, and some antiviral nanomaterials.
- Description of vaccines in Introduction should be expanded. Please cite here the article https://www.thno.org/v11p1690
- "2.2. Coronavirus life cycle" - is described very short, without details. Should be added role of all NSPs in replication and regulation.
- In part "3. Potential therapeutics for management of COVID-19", Authors should add effective doses of presented substances,
- In line 271 is lack of k i NF-kB.
- Why Authors draw only figure for trehalose (Figure 4) and chloroquine (Figure 5), but not also for other substances? Please add all mechanisms.
- "3.10. Ivermectin" is described very poor. Please extend information about ivermectin.
- In Table 1, should be added method of study and dose.
- "4. Conclusion and outlook" is too long. Conclusions should be contained in some (3-5) sentences, not on one page.
Author Response
AUTHORS’ REPLY TO REVIEWERS’ COMMENTS
Title: The molecular basis of COVID-19 pathogenesis, conventional and nanomedicine therapy
Authors: Shirin Kouhpayeh †, Laleh Shariati †, Maryam Boshtam, Ilnaz Rahimmanesh, Mina Mirian, Yasaman Esmaeili, Malihe Najaflu, Negar Khanahmad, Mehrdad Zeinalian, Maria Trovato, Franklin R Tay, Hossein Khanahmad*, and Pooyan Makvandi*
Journal: International journal of Molecular Sciences
Manuscript Reference Number: ijms-1221052
We thank the referees for their useful comments which helped us improve the attached revised manuscript. The manuscript was revised according to the referees’ comments. All changes were highlighted in the revised manuscript as Supporting Information. We hope that our revised manuscript will be acceptable for publication in International journal of Molecular Sciences.
Please find below the replies to the referees’ comments.
…………………………………………………………………………………………………
Point-by-Point Response to the Editors and Reviewer’s Comments.
Referee:3
Comments to the author
1.In Abstract is description of human cell death by coronaviruses. Please, remove it, and write in Abstract what conclusions are from this review, mention therapeutic targets, and some antiviral nanomaterials.
Response: The cell death description was removed, moreover, therapeutic targets and anti-viral nanomaterials were added to abstract
2.Description of vaccines in Introduction should be expanded. Please cite here the article https://www.thno.org/v11p1690
Response: The vaccine description in introduction section was expanded regarding the suggested reference.
3."2.2. Coronavirus life cycle" - is described very short, without details. Should be added role of all NSPs in replication and regulation.
Response: The Coronavirus life was expanded regarding the suggested reference.
4.In part "3. Potential therapeutics for management of COVID-19", Authors should add effective doses of presented substances.
Response: The effective dosage of therapeutics for COVID-19 management were added to each sub section.
5.In line 271 is lack of k i NF-kB.
Response: The k letter was added to NF-B in line 271.
6.Why Authors draw only figure for trehalose (Figure 4) and chloroquine (Figure 5), but not also for other substances? Please add all mechanisms.
Response: Thanks for your attention. Since there are limited information about the molecular mechanisms of other substances such as Tannins, clotrimazole, Losartan and so on, we cannot illustrate a separated figure for each one. Nevertheless, the summary of the molecular mechanisms of these drugs was represented in Figure 8.
- "3.10. Ivermectin" is described very poor. Please extend information about ivermectin.
Response: Thanks for your consideration, indeed, Ivermectin is another name of hydroxychloroquine, therefore, we merged Ivermectin section with hydroxychloroquine section in page 13, line 41-43.
- In Table 1, should be added method of study and dose.
Response: Thanks for your comment. Indeed, this Table is an overview of antiviral mechanisms of nanomaterials against some viruses. And, because this paper focused on the coronaviruses, reporting the nanomaterials’ dose against other viruses is not applicable for this paper. On the other hand, the precise dose of nanomaterials against coronaviruses is not still approve.
9."4. Conclusion and outlook" is too long. Conclusions should be contained in some (3-5) sentences, not on one page.
Response: The conclusion and outlook section was summarized as much as possible.
Round 2
Reviewer 1 Report
The authors responded to all queries. The paper is publishable.
Reviewer 3 Report
Authors significantly corrected manuscript according to reviewer's suggestions. Recently, I recommend article for publication.